# The Influence of Solvent and Extraction Time on Yield and Chemical Selectivity of Cuticular Waxes from *Quercus suber* Leaves

Rita Simões, Isabel Miranda * and Helena Pereira 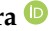

Centro de Estudos Florestais (CEF), Associate Laboratory TERRA, Instituto Superior de Agronomia, Universidade de Lisboa, Tapada da Ajuda, 1349-017 Lisbon, Portugal
* Correspondence: imiranda@isa.ulisboa.pt

**Abstract:** The cuticular lipid compounds, usually named cuticular waxes, present in the cuticular layering of *Quercus suber* adult leaves were extracted with solvents of different polarities (n-hexane, dichloromethane and acetone) and analysed by GC–MS. *Q. suber* leaves have a substantial cuticular wax layer (2.8% of leaf mass and 239 $\mu g/cm^2$), composed predominantly by terpenes (43–63% of all compounds), followed by aliphatic long chain molecules, mainly fatty acids, and by smaller amounts of aliphatic alcohols and n-alkanes. The major identified compound was lupeol (1.2% of leaves in n-hexane extract). The recovery and composition of cuticular lipids depended on the solvent and extraction time. The non-polar or weak polar solvents n-hexane and dichloromethane extracted similar lipid yields (77% and 86% of the total extract, respectively) while acetone solubilised other cellular compounds, namely sugars, with the lipid compounds representing 43% of the total extract. For cuticular lipids extraction, solvents with a low polarity such as n-hexane are the more suitable with an adequate extraction duration, e.g., n-hexane with a minimum extraction of 3 h.

**Keywords:** cork oak; cuticular wax; solvents; terpenes; lupeol

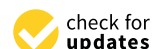



## 1. Introduction

The extracellular surfaces of plant leaves are covered by a layer consisting of cutin, an insoluble polyester glyceride and of a complex mixture of lipids soluble in organic solvents, commonly called cuticular waxes, which are located within and above the cutin structural matrix and named, respectively, intracuticular and epicuticular waxes [1,2]. This hydrophobic interface confers a resistance to a wide range of biotic and abiotic stresses and is involved in the regulation of non-stomatal water loss and gas exchange [2–4]. The functional barrier against water diffusion through the cuticle is preferentially established by intracuticular lipids [5–7] which consist of long-chain aliphatic molecules and alicyclic waxes, including triterpenoids, while the epicuticular lipids, comprising long-chain aliphatic molecules (e.g., alkanes, primary alcohols, fatty acids), are important for interactions between leaf surface and environment [5–7].

The cuticle, as a boundary between the plant and environment, has an enhanced role under adverse environmental conditions. This is the case for most species growing in the Mediterranean region under high solar irradiances, air temperatures, and vapor pressure deficits with a limited water availability, thereby showing a high sclerophyll character. Under drought stress, plants react namely by an ABA-induced stomatal closure, the accumulation of cuticular lipids, and the formation of a deep root system which improves their drought tolerance. Cork oak (*Quercus suber* L.), an evergreen sclerophyllous tree species distributed in the western Mediterranean basin, is an example of a species with great economic importance due to its production of cork, a material with a very interesting set of properties that feeds a dedicated industrial chain [8].

In leaves of *Q. suber* sampled in spring and summer, Martins et al. [9] described a layer of cuticular wax composed mainly of n-alkyl esters (25–45% of the wax extract) and alkanols

(18–50%) with a significant content variability. A more comprehensive study by our group on the cork oak leaves of trees from six seed provenance sources observed that a substantial wax layer (154–235 $\mu g/cm^2$) covered the leaves, with the majority of compounds being pentacyclic triterpenoids and long chain aliphatic compounds, and related the cork oak cuticular characteristics to the species since the trees of different seed origins did not differ [10]. The triterpene fraction contained a large amount of lupeol, a compound that has shown beneficial effects on human health [11–15].

Given the important role of the cuticular waxes in establishing the tree adaptation to adverse environmental conditions, it is of a high interest to clarify the influence of extraction conditions in their content and compositional determinations. In this study, we provide a characterization of the lipophilic compounds present in the leaf cuticle of *Q. suber* extracted with solvents of different polarities and with different extraction times. The results presented here will contribute to the knowledge of the lipophilic compounds in *Q. suber* cuticular leaves and their extraction specificity, allowing to direct the solubilisation process to obtain potential highly valuable phytochemicals of diverse industrial interest.

## 2. Materials and Methods

### 2.1. Plant Material

Mature leaves were collected from two mature *Q. suber* L. trees that were never submitted to cork removal, grown on the campus of the School of Agriculture, in the region of Lisbon, Portugal. The leaves were collected randomly from different branches on the south exposed crown side, in the lower part of the canopy up to a height of approximately 2 m, making up a total sample per tree of about 200 leaves. A composite sample was prepared by combining the leaves of the two trees.

The area of 80 leaves randomly selected from the composite sample was measured by digitalizing and it was calculated with Leica Qwin vs. 3.0 Image Analysis Software.

### 2.2. Cuticular Wax Extraction

The influence of the solvent polarity and extraction time on the yields of solubilized cuticular waxes and on their chemical profile was studied. Conventional Soxhlet extractions were used for the extraction of the cuticular waxes from the whole and intact cork oak leaves taken randomly from the leaf composite sample. Three replicates of leaf samples (without petiole) with a dry mass of around 1.5 g (corresponding to about 10 leaves) were placed in the Soxhlet apparatus and the lipophilic fractions were extracted using organic solvents n-hexane (dielectric constant 2.02), dichloromethane (dielectric constant 9.1), and acetone (dielectric constant 20.7), with different extraction times (5 min, 1, 2, 3, and 6 h with each solvent). The amount of the soluble compounds removed with each solvent and extraction time was determined from the mass difference in the extracted leaves after drying at 105 °C. The results were expressed as a mass percent of the leaf dry mass (g per 100 g leaves) and on a leaf surface area basis ($\mu$g per $cm^2$ leaf area) as the ratio between the extract and the two-sided leaf surface area, obtained by digitalization.

The lipid fraction in the crude extracts was determined from the mass of lipidic compounds estimated by GC–MS (alkanes, alkanols, fatty acids, terpenes, and sterols) and expressed as a percent of the extract mass and as a percent of the leaf dry mass. The estimated quantification of the compounds in the GC–MS chromatograms was done by the calculation of each compound peak area and expressed as a percent of the total chromatogram peak area.

### 2.3. Cuticular Wax Composition

The extracts were analysed using gas chromatography–mass spectrometry (GC–MS) after trimethylsilylation according to Simões et al. [10]. The extract was treated with BSTFA (N,O–bis(trimethylsilyl)trifluoroacetamide) in pyridine (30 min at 70 °C) and analysed in a gas chromatograph coupled to a mass spectrometer (EMIS, Agilent 5973 MSD, Palo Alto, CA, USA) using a Zebron 7HG-G015-02 column (30 m, 0.25 mm; ID, 0.1 $\mu$m film

thickness) with an electron ionization at 70 eV, and helium as a carrier gas at 1.0 mL min$^{-1}$ flux. The column temperature was initially held at 50 °C for 1 min, raised to 150 °C at a rate of 10 °C min$^{-1}$, then to 300 °C at 4 °C min$^{-1}$, to 370 °C at 5 °C min$^{-1}$, and at 8 °C min$^{-1}$ until it reached 380 °C, and followed by an isothermal 5 min period. The compounds were identified and quantified as TMS derivatives by comparing their mass spectra with a GC–MS spectral library (Wiley, NIST) and by comparing their fragmentation profiles with the published data, reference compounds, ion fragmentation patterns, or retention times [10]. The results are given as a percent of each compound peak area to the total chromatogram peak areas. The results are to be considered as only semiquantitative since no internal standards were added nor were the response factors of the different compounds calculated.

The proportion of specific chemical classes in the extracts (e.g., lipids, terpenes, aromatics, sugars), expressed as a % of the extract or in mg per g of leaf dry mass, was calculated using the extract amount per dry leaf mass and the compound area proportion in the total GC–MS chromatogram.

## 3. Results and Discussion

In the present work, the effect of the solvent and extraction time on the yield and composition of the cuticular wax removed from the *Q. suber* leaves surface was studied using three solvents with a different polarity index (by increasing polarity: n-hexane, dichloromethane and acetone) and different extraction times. In the procedure used here, the solubilisation of the compounds refers to both the adaxial and abaxial leaf surfaces. With the relatively long extraction time by Soxhlet (from 1 h to 6 h), the cuticular lipids that are released correspond to a mixture of both the epicuticular and intracuticular lipids [5,16]. Therefore, the resulting extracts reflect the total lipid composition, averaging over the entire depth of the cuticle, rather than assessing only the surface. The potential exception is the 5 min extraction which may target mostly the epicuticular waxes, as discussed later.

Although some techniques have been applied in other works to study separately the chemical composition of epicuticular and intracuticular lipids, e.g., by physically stripping the epicuticular lipids from the leaf surfaces or by using very short dipping times [5,7,17], this was not the objective of the present work. Therefore, the results presented below on the cuticular lipids comprise both epicuticular and intracuticular compounds.

A specific observation should be made on the term cuticular waxes used throughout this work. Although wax is chemically defined as very long-chain esters, the compounds solubilised from the leaf cuticle by the organic solvents have a complex composition comprising long chain fatty acids, primary and secondary alcohols, aldehydes, wax esters, ketones and linear hydrocarbons, that are derived from fatty acid precursors [18]. A more accurate designation should therefore be "extracellular surface lipids" [19]. However, most literature dealing with leaf cuticles uses the term cuticular waxes, and this was maintained here.

### 3.1. Extraction Yields

Table 1 shows the average extract yields (in g per 1000 g of dry leaves) and leaf surface coverage (in µg per cm$^2$ leaf surface) obtained with the different solvents and extraction times. A previous study from our group on the cuticular waxes of cork oak leaves from different tree provenances showed that a 6 h Soxhlet extraction with dichloromethane solubilised on average 2.2% of the dry leaf mass, corresponding to a surface coverage of 189 µg/cm$^2$ [10]. The results obtained here with dichloromethane after 6 h (2.8% of leaf mass and 239 µg/cm$^2$) are similar to those previously reported.

There was a clear difference between the solvents in their extracting ability: the yields were higher for acetone and hexane (3.6% and 3.4% of the leaf mass, respectively, after 6 h of extraction) in comparison to dichloromethane. The extraction time also had an effect on the extract yield (Table 1). Increasing the extraction time resulted in a higher removal, although the relative extraction power of the three solvents was maintained, i.e., lower for

dichloromethane. It should be noted that the extraction yields are related to the solubilization power and selectivity of the solvent to both lipid and non-lipid compounds, as given namely by their polarity, and discussed in detail in the following sections.

**Table 1.** Yields of material solubilized from the surface of whole *Q. suber* leaves by different solvents and extraction times, in g per 100 g dry leaves, and µg per cm$^2$ of leaf surface.

|  | 5 min | 1 h | 2 h | 3 h | 6 h |
|---|---|---|---|---|---|
| **n-Hexane** |  |  |  |  |  |
| g/100 g leaves | 0.33 ± 0.04 | 1.92 ± 0.72 | 1.98 ± 0.25 | 3.11 ± 0.11 | 3.39 ± 0.19 |
| µg/cm$^2$ | 28.13 ± 3.09 | 165.67 ± 62.33 | 170.58 ± 21.52 | 286.02 ± 9.27 | 287.15 ± 23.18 |
| **Dichloromethane** |  |  |  |  |  |
| g/100 g leaves | 0.51 ± 0.03 | 1.17 ± 0.09 | 1.23 ± 0.13 | 1.79 ± 0.33 | 2.77 ± 0.17 |
| µg/cm$^2$ | 43.83 ± 2.88 | 101.36 ± 7.66 | 108.30 ± 11.17 | 154.10 ± 28.52 | 239.00 ± 14.25 |
| **Acetone** |  |  |  |  |  |
| g/100 g leaves | 0.35 ± 0.07 | 1.27 ± 0.25 | 2.70 ± 0.15 | 3.16 ± 0.02 | 3.57 ± 0.49 |
| µg/cm$^2$ | 30.29 ± 5.84 | 109.42 ± 21.81 | 232.86 ± 12.81 | 272.24 ± 2.12 | 308.33 ± 42.32 |

The short 5 min extraction time led to a very partial removal of the soluble compounds, e.g., hexane, dichloromethane, and acetone solubilised only, respectively, 10%, 18%, and 10% of the extracts obtained with a 6 h extraction.

These results point out that a full, or at least a major, solubilisation of the leaf cuticular compounds is only possible if sufficient extraction time is given. Therefore, protocols that use a short or very short contact between the solvent and the leaf surface will only solubilise a small fraction of the cuticular lipids, namely only the epicuticular waxes, as previously discussed [10]. This is the reason for the difference in the extraction yield found between the present results and a previous study on young cork oak leaves using a quick dipping and shaking in the chloroform [9].

*3.2. Effect of Solvent on Lipid Extraction*

The extraction yield, i.e., the compounds solubilized by the solvent in relation to the leaf mass, is a quantitative but undifferentiating indicator of the molecules removed from the surface of the leaves, meaning that other non-lipid compounds may be solubilized, and more so with solvents with an increased polarity. Previous reports on the composition of the solubilized compounds from *Q. suber* leaves using dichloromethane showed that 80% of the extract was of a lipid nature, with about 2% corresponding to aromatics and a small proportion of sugars, as calculated from the corresponding peak areas in relation to the total chromatogram peak area [10].

The composition of the extracts is therefore an important aspect when analysing the solvent efficiency to solubilize the cuticular lipids. Considering the total amount of lipidic compounds present in the cuticular extracts, as determined from their GC–MS analysis (which will be reported in detail in the next section), by grouping alkanes, alkanols, fatty acids and glycerides, terpenes, and sterols, a new insight is obtained on the solvent role to attain higher lipid yields. Figure 1 represents the lipid yields obtained with the three solvents and five extraction times. The proportion of the lipid extractives was consistent with the order of the polarity of the solvent.

The 6 h acetone extraction shows the lowest proportion of extracted lipid compounds as estimated by their peak area proportion in the total chromatogram (15.3 mg/g of dry leaves, corresponding to 43% of the total extract), while the non-polar or weak polar solvents n-hexane and dichloromethane extracted a similar lipid proportion estimated as 26.1 and 24.1 mg/g of dry leaves, corresponding to 77% and 86% of the total extract, respectively (Figure 1). The lipid proportion in the extracts increased with the extraction time, e.g., from 1.7% to 2.6% with n-hexane, and from 0.5% to 1.5% with acetone for 1 h and 6 h extractions, respectively.

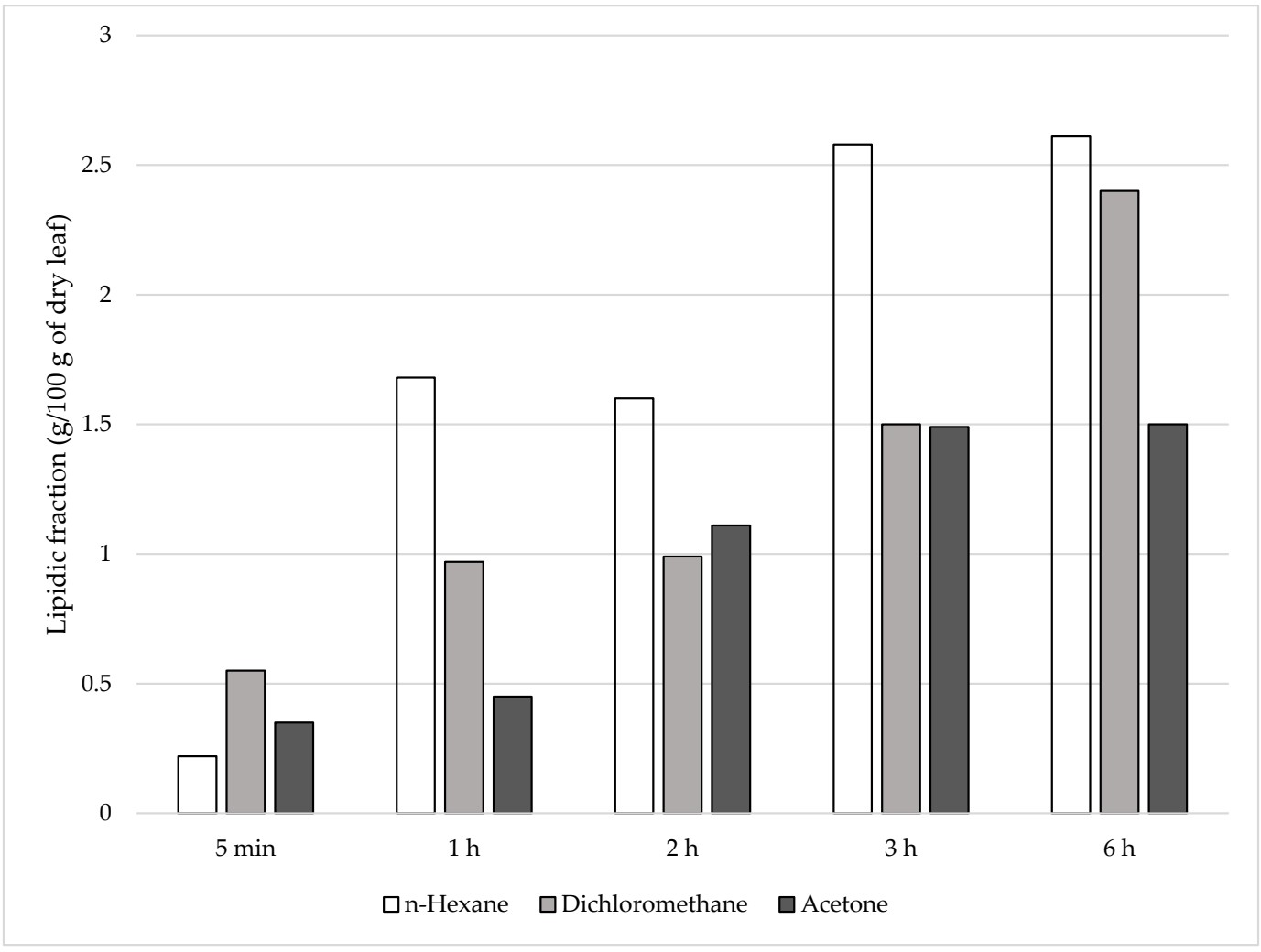

**Figure 1.** Lipidic fraction obtained from extraction of *Q. suber* leaves with solvents with different polarity indices with different extraction times.

For solubilisation of the cuticular lipids from *Q. suber* leaves (Figure 1), n-hexane was the most effective solvent in providing the highest recovery, which was attained with a 3 h extraction in the experimental conditions used. Dichloromethane, as used in the previous *Q. suber* cuticular studies, also provided a similar lipid recovery with a 6 h extraction [10].

### 3.3. Compositional Analysis of Extracts by Chemical Family

The composition by the chemical family of the cuticular waxes of cork oak leaves solubilised by different solvents with different extraction times, as a percentage of the total peak area in the GC–MS chromatograms, are presented in Table 2, and detailed for the three solvents in Tables 3–5. The results are a semiquantitative approximation of compound proportion since no internal standards were added or response factors were calculated.

The composition of n-hexane and dichloromethane extracts is quite similar. With the exception of the 5 min extracts, that will be discussed subsequently, the extracts are mainly made up of terpenes (43–63% of all compounds), followed by aliphatic long chain molecules, mainly fatty acids (11–26% of all compounds), and by smaller amounts of aliphatic alcohols, n-alkanes, and with some phenolics. This is in accordance with the reported composition of the cuticular wax from *Q. suber* leaves where triterpenes and aliphatic compounds predominate (61–72% and 17–23% of the total compounds, respectively) [10]. For the short extraction time of 5 min, the compounds solubilised were aliphatic alcohols that represented

98% and 96% of all compounds, respectively, in the n-hexane and dichloromethane extracts. This suggests that the more easily accessible external layer of the cuticle predominantly contains alkanols [20].

**Table 2.** Composition of the cuticular extracts of leaves from *Q. suber* solubilised by different solvents and extraction times by chemical class as a percentage of total peak area in the GC–MS chromatograms.

| | Proportion of Total Compounds, % | | | | | | | | | | | | | |
|---|---|---|---|---|---|---|---|---|---|---|---|---|---|---|
| | n-Hexane | | | | | Dichloromethane | | | | | Acetone | | | | |
| Family | 5 min | 1 h | 2 h | 3 h | 6 h | 5 min | 1 h | 2 h | 3 h | 6 h | 5 min | 1 h | 2 h | 3 h | 6 h |
| Alkanes | 0.3 | 6.9 | 3.5 | 6.9 | 6.6 | 0.4 | 3.4 | 3.1 | 2.2 | 3.6 | - | 0.5 | 1.8 | 0.4 | 0.2 |
| Alkanols | 98.4 | 5.5 | 4.0 | 3.7 | 7.0 | 95.5 | 5.1 | 7.3 | 5.5 | 7.3 | 2.1 | 3.8 | 3.7 | 4.2 | 2.0 |
| Fatty acids | 0.2 | 11.8 | 11.0 | 16.3 | 16.1 | 0.9 | 12.6 | 25.0 | 25.3 | 20.0 | 39.6 | 5.4 | 11.6 | 11.7 | 9.0 |
| Glycerides | - | 0.1 | 0.2 | 0.5 | 0.3 | - | 1.0 | 2.5 | 0.9 | 0.2 | - | 1.1 | 1.9 | 1.8 | 1.2 |
| Sterols | - | 0.01 | 3.2 | 4.2 | 2.7 | - | 2.0 | 3.0 | 2.7 | 3.4 | - | 1.1 | 1.5 | 1.6 | 1.2 |
| Terpenes | 0.1 | 59.7 | 63.4 | 52.0 | 44.6 | 2.0 | 59.9 | 42.7 | 48.4 | 44.8 | - | 19.5 | 20.3 | 20.5 | 22.2 |
| Aromatics | - | 1.7 | 4.5 | 5.6 | 4.1 | - | 4.7 | 5.6 | 5.7 | 4.5 | 13.5 | 2.9 | 3.2 | 2.6 | 2.7 |
| Sugars | - | 0.0 | 0.0 | 0.0 | 0.0 | - | 0.1 | 0.5 | 1.4 | 0.6 | 39.7 | 56.8 | 50.0 | 41.3 | 46.8 |
| Others | - | 1.6 | 0.9 | 0.6 | 0.3 | - | 1.1 | 0.5 | 1.4 | 1.4 | 5.0 | 6.2 | 1.7 | 9.9 | 8.4 |
| Total Identified | 99.0 | 87.3 | 90.7 | 89.8 | 81.7 | 99.5 | 90.0 | 90.7 | 93.7 | 85.5 | 99.8 | 93.4 | 92.0 | 89.7 | 91.6 |

**Table 3.** Composition of the cuticular waxes of cork oak leaves solubilised by n-hexane with different extraction times, as determined by GC–MS, in % of total peak area (only compounds with over 1% are shown; complete composition in Supplementary Table S1).

| | Extraction Time | | | | |
|---|---|---|---|---|---|
| Wax Constituent (% Total Peak Area) | 5 min | 1 h | 2 h | 3 h | 6 h |
| **Alkanes** | | | | | |
| Heptacosane ($C_{27}$) | - | - | - | 1.12 | 1.02 |
| Triacontane ($C_{30}$) | - | 5.10 | 2.47 | 4.89 | 4.66 |
| **Alkanols** | | | | | |
| Hexadecan-1-ol ($C_{16}OH$) | - | 1.15 | - | - | - |
| Docosan-1-ol ($C_{22}OH$) | 1.24 | - | - | - | - |
| Tretracosan-1-ol ($C_{24}OH$) | 96.64 | 2.91 | 3.11 | 3.05 | 4.34 |
| **Fatty acids** | | | | | |
| **Saturated** | | | | | |
| Hexadecanoic acid ($C_{16:0}$) | - | 7.57 | 5.35 | 5.57 | 5.32 |
| Hexacosanoic acid ($C_{26:0}$) | - | - | - | 1.00 | 1.05 |
| Octacosanoic acid ($C_{28:0}$) | - | - | - | 2.61 | 3.39 |
| Triacontanoic acid ($C_{30:0}$) | - | - | - | 1.08 | 1.37 |
| **Unsaturated** | | | | | |
| 9,12-Octadecadienoic acid ($C_{18:2}$) | - | - | - | 1.12 | 0.81 |
| 9,12,15-Octadecatrienoic acid ($C_{18:3}$) | - | - | 2.28 | 3.43 | 2.11 |
| **Sterols** | | | | | |
| β-Systosterol | - | - | 2.82 | 3.50 | 2.50 |
| **Terpenes** | | | | | |
| **pentacyclic triterpenes** | | | | | |
| β-Amyrin | - | 3.02 | 2.83 | 3.11 | 2.47 |
| Germanicol | - | 11.19 | 17.31 | 7.49 | 6.00 |
| Lupeol | - | 43.93 | 40.43 | 37.76 | 33.04 |
| Betulin | - | - | - | 1.12 | 0.97 |
| **Aromatics** | | | | | |
| Hexadecyl-(E)-p-coumarate | - | - | 4.21 | 5.32 | 3.66 |

**Table 4.** Composition of the cuticular waxes of cork oak leaves solubilised in dichloromethane with different extraction times, as determined by GC–MS in % of total peak area (only compounds with over 1% are shown; complete composition in Supplementary Table S2).

| Wax Constituent (% Total Peak Area) | Extraction Time | | | | |
| --- | --- | --- | --- | --- | --- |
| | 5 min | 1 h | 2 h | 3 h | 6 h |
| Alkanes | | | | | |
| Nonacosane ($C_{29}$) | - | 2.53 | 2.18 | 1.47 | 2.56 |
| Alkanols | | | | | |
| Hexadecan-1-ol ($C_{16}OH$) | 0.90 | - | - | - | - |
| Docosan-1-ol ($C_{22}OH$) | 2.38 | 0.77 | 1.12 | 1.10 | 1.85 |
| Tretracosan-1-ol ($C_{24}OH$) | 92.12 | 3.47 | 4.59 | 3.31 | 4.62 |
| Fatty acids | | | | | |
| Saturated | | | | | |
| Hexadecanoic acid ($C_{16:0}$) | - | 7.60 | 9.84 | 9.71 | 9.68 |
| Octacosanoic acid ($C_{28:0}$) | - | 1.60 | 2.31 | 1.14 | 2.49 |
| Unsaturated | | | | | |
| 9,12-Octadecadienoic acid ($C_{18:2}$) | - | 0.14 | 1.73 | 2.38 | 0.75 |
| 9,12,15-Octadecatrienoic acid ($C_{18:3}$) | - | 0.39 | 6.43 | 8.04 | 2.69 |
| Glycerides | | | | | |
| Glycerol | - | 0.95 | 2.14 | 0.51 | - |
| Sterols | | | | | |
| β-Systosterol | - | 1.59 | 2.55 | 2.33 | 3.24 |
| Terpenes | | | | | |
| Pentacyclic triterpenes | | | | | |
| β-Amyrin | - | 2.75 | 2.29 | 2.56 | 2.11 |
| Germanicol | - | 9.15 | 5.46 | 6.46 | 4.77 |
| Lupeol | 1.61 | 45.44 | 31.56 | 35.28 | 35.37 |
| Aromatics | | | | | |
| Hexadecyl-(E)-p-coumarate | - | 4.27 | 5.31 | 5.19 | 4.09 |
| Sugars | | | | | |
| Fructofuranose | - | 0.09 | 0.54 | 1.43 | 0.18 |

**Table 5.** Composition of the cuticular waxes of cork oak leaves solubilised in acetone with different extraction times, as determined by GC–MS, in % of total peak area (only compounds with over 1% are shown; complete composition in Supplementary Table S3).

| Wax Constituent (% Total Peak Area) | Extraction Time | | | | |
| --- | --- | --- | --- | --- | --- |
| | 5 min | 1 h | 2 h | 3 h | 6 h |
| Alkanols | | | | | |
| 1,4-Butanediol | 1.85 | - | - | - | - |
| Tretracosan-1-ol ($C_{24}OH$) | - | 3.22 | 3.21 | 2.74 | 1.28 |
| Fatty acids | | | | | |
| Saturated | | | | | |
| Hexadecanoic acid ($C_{16:0}$) | - | 2.51 | 5.54 | 5.92 | 4.35 |
| Unsaturated | | | | | |
| 9,12-Octadecadienoic acid ($C_{18:2}$) | - | 0.03 | 0.66 | 1.23 | 0.48 |
| 9,12,15-Octadecatrienoic acid ($C_{18:3}$) | - | 0.07 | 0.61 | 0.07 | 1.26 |
| Diacids | | | | | |
| Butanedioic acid ($C_{4:0}$) | - | 0.95 | 0.99 | 2.45 | 1.69 |
| Glycerides | | | | | |
| Glycerol | - | 1.04 | 1.72 | 1.36 | 0.95 |
| Sterols | | | | | |
| β-Systosterol | - | 0.32 | 0.99 | 1.43 | 1.15 |

**Table 5.** *Cont.*

| Wax Constituent (% Total Peak Area) | Extraction Time | | | | |
|---|---|---|---|---|---|
| | **5 min** | **1 h** | **2 h** | **3 h** | **6 h** |
| Terpenes | | | | | |
| pentacyclic triterpenes | | | | | |
| β-Amyrin | - | 0.72 | - | 1.18 | 1.19 |
| Germanicol | - | 2.89 | 2.05 | 2.71 | 2.86 |
| Lupeol | - | 14.53 | 16.25 | 14.71 | 16.15 |
| Aromatics | | | | | |
| Catechine | 13.18 | - | - | - | - |
| Hexadecyl-(E)-p-coumarate | | 1.67 | 1.62 | 2.35 | 2.42 |
| Sugars | | | | | |
| Myo Inositol | 19.98 | - | 15.34 | - | −0.11 |
| Scyllo Inositol | - | 15.44 | 14.27 | 12.39 | 14.88 |
| Deoxinositol | - | 6.28 | 1.81 | 5.24 | 6.3 |
| D-Fructose | - | 15.47 | 11.27 | 10.68 | 8.61 |
| (α/β) D-Glucopyranose (isomer) | - | 17.11 | 6.28 | 11.18 | 15.01 |
| Sucrose | 18.62 | 0.48 | 0.21 | 0.47 | 0.4 |
| Others compounds | | | | | |
| Quininic acid | 38.18 | 4.6 | 0.94 | 7.2 | 7.15 |
| Shikimic acid, 4TMS derivative | 5.03 | - | - | - | - |

The compositional profile of the acetone extract clearly differs, with sugars being the main chemical class (57–47% of all compounds), followed by triterpenes (19–23%), and fatty acids (5–12%) with lower proportions of the other chemical families. The presence of other non-lipid compounds, mainly of sugars, indicates that, in contrast to the other solvents used, acetone penetrates into the leaf interior and solubilises the more polar sugar molecules. The degradation of the cellular membranes by solvents such as chloroform and acetone have been reported, leading to the solubilisation of cellular compounds not related to the cuticular matrix, for instance of indole alkaloids [21], and chlorophyll and cytoplasmic sterols. The effect of acetone on the leaf cells was noticed already for the short 5 min extraction with the solubilisation of fatty acids, sugars, and aromatics. This is in agreement with reports of an instant destruction of cell integrity when using chloroform [21].

The recovery (in g/100 g leaves) of the extracted cuticular lipids by the chemical family under the different conditions is shown in Figure 2. The difference in the acetone extract in comparison with the hexane and dichloromethane extracts is clear with the major extract fraction being sugars representing 1.67 g/100 g leaves (6 h extraction). Hexane and dichloromethane permit the recovery of cuticular terpenes (1.51 and 1.24 g/100 g leaves, respectively) and fatty acids (0.55 and 0.55 g/100 g leaves, respectively).

The extraction time had only a moderate impact on the recovery of the extracellular cuticular lipids with the differences given more by the extraction yield (Table 1) than by the compositional profile, which remains independent of the extraction time (Table 2). For instance, the recovery of terpenes with n-hexane increased from 11.5 mg/g of leaves with 1 h to 15.1 mg/g of leaves with 6 h of contact, while the extraction with dichloromethane was the one most impacted by duration, e.g., terpenes recovery increased from 7.0 mg/g of leaves with 1 h to 12.4 mg/g of leaf with 6 h of contact (Figure 2).

These results should be taken with caution and restricted to the experimental procedure used here (Soxhlet extraction). Martins et al. [9] studied the extraction of extracellular surface lipids in the leaves of young *Q. suber* by immersion for a few seconds in chloroform, thus extracting only the epicuticular layer, and this explains the compositional differences in relation to our results, for example, the extract contained only small amounts of triterpenoids. Loneman et al. [19] applied a quick surface extraction method by dipping maize seedling leaves with chloroform and hexane:diethyl ether for 1 to 10 min, and observed that the recovery and composition depended on the solvent and duration of the extraction.

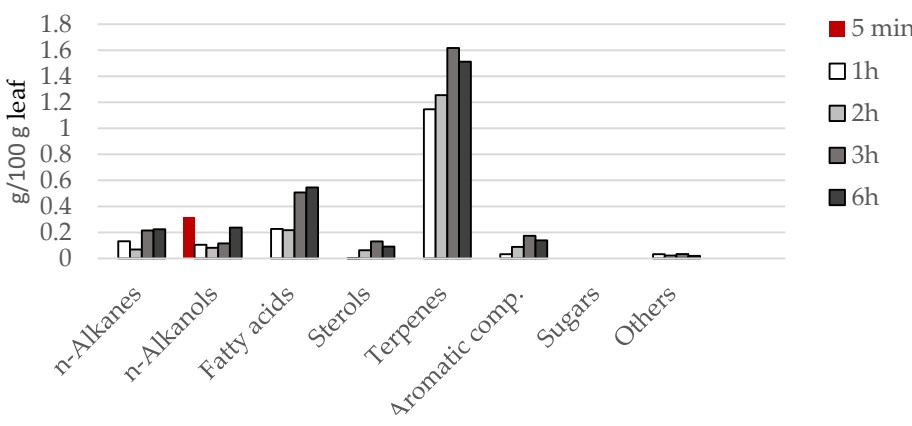

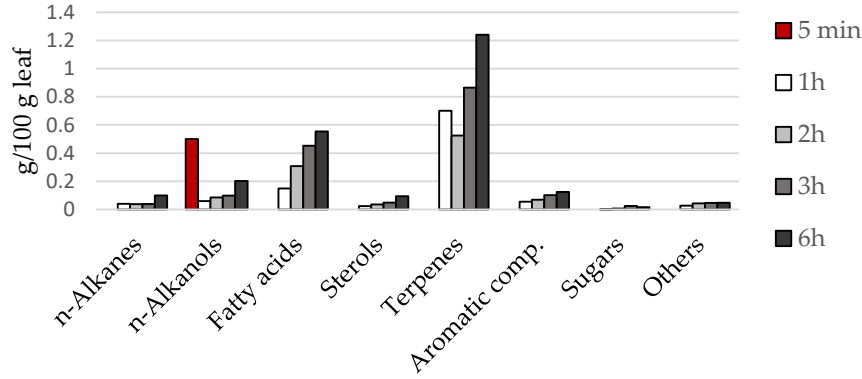

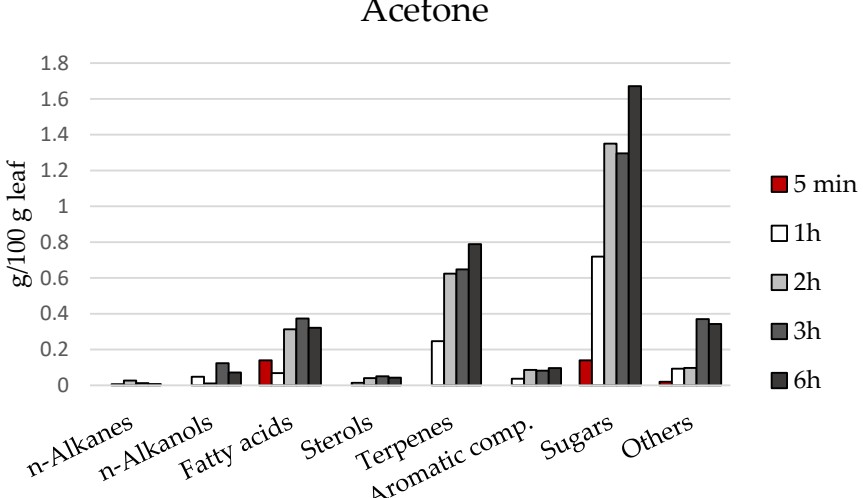

**Figure 2.** Composition (g/100 g leaves) by chemical family of the cuticular lipids extracted from *Q. suber* leaves with different solvents and extraction time.

*3.4. Chemical Composition of Extracts*

The composition of n-hexane, dichloromethane, and acetone extracts is summarised in Tables 3–5 that include only the compounds with at least 1.0% of the chromatogram peak area. The complete compositional profile and the GC–MS chromatogram of the derivatised extracts at 5 min and 3 h is given as the supplementary data (Supplementary Tables S1–S3, respectively,

for n-hexane, dichloromethane, and acetone extracts) and Supplementary Figures S1–S3, respectively, for n-hexane, dichloromethane, and acetone extracts at 5 min and 3 h, respectively, exemplify the quality of the chromatograms that were obtained for all the cases.

In the terpenic family, the triterpenoid lupeol was among the most prominent individual compound, followed by germanicol and α- and β-amyrin, e.g., lupeol with 33–43% of all compounds (about 75% of the terpenic fraction) and germanicol with 6–17% of all compounds (about 15% of terpenic fraction). Smaller amounts of other triterpenoids such as betulin, betulinic, oleanolic, and ursonic acids were also identified (Supplementary Tables S1–S3). All of the identified triterpenes were previously reported as cuticular wax components of *Q. suber* leaves [10].

Triterpenoids are one of the largest and structurally diverse classes of natural specialized metabolites with various ecological and biological properties, with a broad spectrum of relevant clinical activities, such as antihypertensive, antidiabetic, anti-inflammatory, anticancer, and antimicrobial properties leaves [11–15]. The high proportion of lupeol in the cuticular extracts may lead to considering *Q. suber* leaves as a source of this compound, e.g., a lupeol recovery of 1.18 g/100 g of leaves may be obtained with a 3 h n-hexane extraction.

Four sterols were identified in the extracts (β-systosterol, δ-tocopherol; stigmasterol and 25-hydroxycholesterol, Supplementary Tables S1–S3) with β-systosterol showing the highest abundance with 2.5–3.5% of all compounds.

The hydrocarbon fraction includes fatty acids, n-alkanols, and n-alkanes. With 6 h of contact, fatty acids are mostly saturated fatty acids, accounting to 79% and 59% of the total fatty acids in the n-hexane and dichloromethane extracts, and 59% of the total fatty acids content in the acetone extract. Fatty acids were identified in the homologous series from heptanoic acid (C7:0) to triacontanoic acid (C30:0), with a strong even-over-odd carbon atom predominance. Hexadecanoic acid (palmitic acid, C16:0) was the most abundant saturated fatty acid in the three extracts (5.3% of all compounds in n-hexane extract, 9.7% in dichloromethane extract, and 4.4% in acetone extract). Smaller amounts of octacosanoic acid (C28:0), hexacosanoic acid (C26:0), and triacontanoic acid (C30:0) were also found. Three unsaturated acids were identified in the extracts: 9,12,15-octadecatrienoic (linoleic acid), 9,12-octadecadienoic (linolenic acid), and 9-cis-hexadecenoic (palmitoleic acid) acids.

Fatty alcohols represented only a small fraction of the total amount of lipophilic extractives. A series of n-fatty alcohols ranging from C4 to C28 were present in cork oak wax leaves extracts, with tetracosanol (C24OH) as the most abundant (4.3% of all compounds in the n-hexane wax extract, 4.6% in the dichloromethane extract, and 1.4% in the acetone wax extract)

A series of n-alkanes ranging from nonadecane (C19) to hentriacontane (C31) occurred in the extracts, with a strong odd-over-even carbon atom number predominance. Nonacosane (C29) was the most abundant in the dichloromethane extract (72% of alkanes) and triacontane (C30 in the n-hexane extract (74% of alkanes)).

The composition of the identified aliphatic compounds is similar to that obtained for the cuticular wax of *Q. suber* leaves from different sources extracted with dichloromethane with 6 h of contact, that included fatty acids with a chain length C30, C28, and C16, n-alkanes with chain lengths ranging between C15 and C31 [10].

Low amounts of acylglycerols including mono-glycerides such as monopalmitin, 2-monostearin, 1-linolenoylglycerol, and 3-hydroxypropyl hexadecanoate were identified in all the extracts, together representing 0.45%, 0.57%, and 0.19% of the n-hexane, dichloromethane, and acetone extracts, respectively.

Aromatic compounds were also detected in all the extracts such as benzoic acid and p-coumarates. P-coumarates were the dominant aromatic compound corresponding to 5.3%, 4.1%, and 2.4% of the n-hexane, dichloromethane, and acetone extracts, respectively.

High amounts of sugars were found in the acetone extracts (56% of all compounds) including fructose, glucose, sucrose, the free cyclitols myo-inositol, scyllo-inositol, and deoxinositol. Scyllo-inositol has been considered as a potential therapeutic agent for Aβ amyloid disorders such as Alzheimer's disease, Down's syndrome, and cerebrovascular dementia [22,23].

## 4. Conclusions

This study confirmed that the extent and selectivity of the solubilisation of non-polar and amphipathic molecules that are part of the cuticular layering of plant leaves, as exemplified for the case of *Q. suber* adult leaves, depends on the solvent and on the extraction duration. Less polar solvents such as n-hexane or dichloromethane solubilize the lipid components and a small number of aromatics, while more polar solvents, such as acetone, remove a large proportion of sugars arising from the solubilisation of cellular material. For the purpose of cuticular lipids extraction, solvents with a low polarity, such as n-hexane, are the more suitable and an adequate extraction duration should be considered for an extensive removal, including both the epicuticular and intracuticular lipids. In the present case, n-hexane with a minimum 3 h extraction is proposed. Although not specifically studied here, the procedural method for the extraction should also play a role in the extent of solubilization. The cuticular lipid compounds, usually named cuticular waxes, include as major components terpenes and a large fraction of long-chain fatty acids and other long-chain aliphatics, with a low proportion of sterols, aromatics, and other compounds. It is therefore probable that the terpenes will establish the transpiration barrier in association with the long chain aliphatic compounds.

**Supplementary Materials:** The following supporting information can be downloaded at: https://www.mdpi.com/article/10.3390/pr10112270/s1, Table S1: Composition of the cuticular waxes of cork oak leaves solubilised in n-hexane with different extraction times as determined by GC–MS, in % of total peak área; Table S2: Composition of the cuticular waxes of cork oak leaves solubilised in dichloromethane with different extraction times as determined by GC–MS, in % of total peak area; Table S3: Composition of the cuticular waxes of cork oak leaves solubilised in acetone with different extraction times as determined by GC–MS, in % of total peak area. Figure S1: GC–MS chromatogram of the derivatised n-hexane extract from the cuticular waxes of cork oak leaves. Figure S2: GC–MS chromatogram of the derivatised dichloromethane extract from the cuticular waxes of cork oak leaves; Figure S3: GC–MS chromatogram of the derivatised acetone extract from the cuticular waxes of cork oak leaves

**Author Contributions:** Conceptualization, H.P. and I.M.; methodology, I.M. and R.S.; validation, H.P., I.M. and R.S.; formal analysis, R.S.; investigation, R.S.; data curation, R.S., H.P. and I.M.; writing—original draft preparation, I.M. and R.S.; writing—review and editing, H.P., I.M. and R.S. All authors have read and agreed to the published version of the manuscript.

**Funding:** This research was funded by Fundação para a Ciência e a Tecnologia (FCT) through funding of the Forest Research Centre (UIDB/00239/2020). Funding for this work was also provided by a doctoral scholarship from FCT SUSFOR Doctoral Programme (PD/BD/128259/2016).

**Data Availability Statement:** Not applicable.

**Acknowledgments:** Rita Simões acknowledges a doctoral scholarship from FCT with the SUSFOR Doctoral Programme (PD/BD/128259/2016). We thank Ana Rodrigues for her technical assistance in field sampling.

**Conflicts of Interest:** The authors declare no conflict of interest.

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
