# Peer review of "The Influence of Solvent and Extraction Time on Yield and Chemical Selectivity of Cuticular Waxes from Quercus suber Leaves"

_processes, doi:10.3390/pr10112270_

Round 1

Reviewer 1 Report

Manuscript is original, well defined, easy to understand. The language is appropriate and understandable. The topic is compatible with the journal’s scope. The results are very significant and relevant, presented in a well-structured manner. The manuscript’s results are reproducible based on the details given in the methods section. The figures (1-2) and tables (1-5) are appropriate, they are clearly presented. Conclusions justified and supported by the results, consistent with the evidence and arguments presented.

Accept manuscript with minor changes:

Line 65: It is not necessary to re-state the full name of the species Quercus suber that was first stated in line 42, but only state the shortened Q. suber.

Line 318: Same as line 65.

Line 394: Remove the parentheses in front of the manuscript title.

Line 354 to line 402: Remove hyperlink where it need.

Line 397: Bold according to Journal Citation Guide for Processes journal.

Line 399: Same as line 397.

Author Response

Dear Reviewer

The authors thank the reviewer for the helpful suggestions that were followed in the present manuscript revision. The response to the specific comments is detailed beneath.

Best regards

Isabel Miranda

Reviewer 2 Report

The paper "The influence of solvent and extraction time on yield and chemical selectivity of cuticular waxes from Quercus suber leaves" examines the efficacy of three solvents in extracting total lipids from Quercus s leaves. The results have been well-explained in the publication, but little is said about how the data were interpreted and analyzed.

There are a few remarks and suggestions. The work can be accepted for publication once they have been addressed.

3.1. Extraction results: comments

Can authors explain why there was a greater mass loss from the surface of the leaves in n-hexane than in DCM? For 6 hours of extraction duration, n-hexane and acetone extracts are practically comparable. Why? When acetone was used instead of n-hexane, did the chemicals that were extracted differ?

What impact did the length of the extraction process have on the chromatogram quality? Did the authors note a significant shift in the background chemical composition? Did they find that using acetone as opposed to n-hexane or DCM increased the extraction of polar compounds? Was there any fronting or tailing at all?

Why wasn't chloroform considered in this research of comparisons? Samples were shaken and extracted in chloroform in the prior investigation in ref. 9. It might have produced greater results than the solvents under evaluation, or vice versa, had it been included for a longer period of extraction in this study.

3.2 Effect of solvent on lipid extraction: Comments

Please explain how the compound yield was calculated.

Please include the chromatograms of the extracts after 5 minutes, 1, 2, 3, and 6 hours of extraction in acetone, DCM, and n-hexane.

If possible, create an extracted ion chromatogram for each compound and label the peaks for each compound detected. Please include this information in the supplementary materials.

What made BSTFA the reagent of choice for derivatization? In addition to masking volatile compounds, BSTFA is known to cause several chromatogram artifacts.

Author Response

(The authors gave the same response as above.)

Round 2

Reviewer 2 Report

Please see my comments in the attached PDF file. There are still some comments that have not been addressed properly. 

In a comment, it was mentioned to put chromatograms of all the extraction aliquots at 5 min to 6 h. But, in supplementary information chromatogram for only 3h extraction analysis is present. 

Which internal standard was used for quantitation work? 

Author Response

Dear Reviewer

The authors thank the reviewer for the suggestions. The response to the specific comments is detailed beneath.

Best regards

Isabel Miranda

Round 3

Reviewer 2 Report

The manuscript's quality has improved since it was first submitted. However, excluding data from peer review because it is not required defeats the purpose of peer review.

It has been agreed that all data cannot be included in the manuscript, but it can be submitted separately for peer review.

Author Response

Dear Reviewer

The response to the specific comments is detailed beneath.

Best regards

Isabel Miranda
